# ADAPTIVE MEMORY NETWORKS

## ABSTRACT

Real-world Question Answering (QA) tasks often consist of thousands of words that represent many facts and entities. Existing models based on LSTMs require a large number of parameters to support external memory and do not generalize efficiently for long sequence inputs. Memory networks address these limitations by storing information to an external memory module but must examine all inputs in the memory. Hence, for longer sequence inputs, the intermediate memory components proportionally scale in size, resulting in poor inference times. We present Adaptive Memory Networks (AMN) that process input-question pairs to dynamically construct a network architecture optimized for lower inference times. AMN creates multiple *memory banks* that store entities from the input story to answer the questions. The model learns to *reason* important entities from the input text based on the question and concentrates these entities within a single memory bank. At inference, one or few banks are used, creating a tradeoff between accuracy and performance. AMN is enabled by first, a novel bank controller that makes discrete decisions with high accuracy and second, the capabilities of a dynamic framework (such as PyTorch) that allow for dynamic network sizing and efficient variable mini-batching. In our results, we demonstrate that our model learns to construct a varying number of memory banks based on task complexity and achieves faster inference times for standard bAbI tasks, and modified bAbI tasks. We solve all bAbI tasks with an average of 48% fewer entities on tasks containing excess, unrelated information.

## 1 INTRODUCTION

Question Answering (QA) tasks are gaining significance due to their widespread applicability to recent commercial applications such as chatbots, voice assistants and even medical diagnosis (Goodwin & Harabagiu (2016)). Furthermore, many existing natural language tasks can also be re-phrased as QA tasks. Providing faster inference times for QA tasks is crucial. Consumer device based question-answer services have hard timeouts for answering questions. For example, Amazon Alexa, a popular QA voice assistant, allows developers to extend the QA capabilities by adding new "Skills" as remote services (Amazon (2017)). However, these service APIs are wrapped around hard-timeouts of 8 seconds which includes the time to transliterate the question to text on Amazon's servers and the round-trip transfer time of question and the answer from the remote service, and sending the response back to the device. Furthermore, developers are encouraged to provide a list of questions ("utterances") apriori at each processing step to assist QA processing (Amazon (2017)).

Modeling QA tasks with LSTMs can be computationally expensive which is undesirable especially during inference. Memory networks, a class of deep networks with explicit addressable memory, have recently been used to achieve state of the art results on many QA tasks. Unlike LSTMs, where the number of parameters grows exponentially with the size of memory, memory networks are comparably parameter efficient and can learn over longer input sequences. However, they often require accessing all intermediate memory to answer a question. Furthermore, using focus of attention over the intermediate state using a list of questions does not address this problem. Soft attention based models compute a softmax over all states and hard attention models are not differentiable and can be difficult to train over a large state space. Previous work on improving inference over memory networks has focused on using unsupervised clustering methods to reduce the search space (Chandar et al. (2016); Rae et al. (2016)). Here, the memory importance is not *learned* and the performance of nearest-neighbor style algorithms is often comparable to a softmax operation over memories.

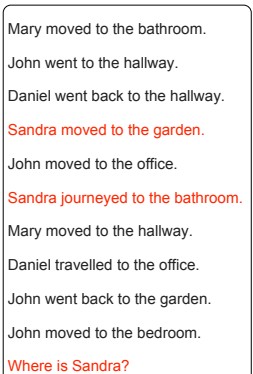 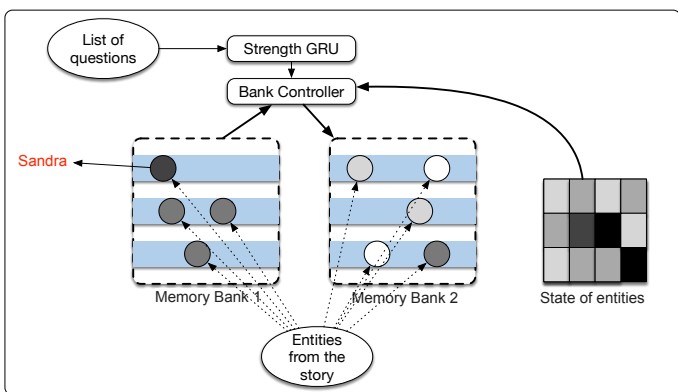

Figure 1: **Overview of Adaptive memory networks. Multiple memory banks are created based on the story and input entities are moved in them based on their relevance to the question. Inference is performed on a single (or less than all) banks most relevant to the question(s).**

To provide faster inference for long sequence-based inputs, we present Adaptive Memory Networks (AMN), that constructs a memory network on-the-fly based on the input. Like past approaches to addressing external memory, AMN constructs the memory nodes dynamically. However, distinct from past approaches, AMN constructs a memory architecture with network properties that are decided dynamically based on the input story. Given a list of possible questions, our model computes and stores the entities from the input story in a *memory bank*. The entities represent the hidden state of each word in the story while a memory bank is a collection of entities that are similar w.r.t the question. As the number of entities grow, our network learns to construct new memory banks and copies entities that are more relevant towards a single bank. Entities may reside in different bank depending on their distance from the question. Hence, by limiting the decoding step to a dynamic number of constructed memory banks, AMN achieves lower inference times. AMN is an end-to-end trained model with dynamic learned parameters for memory bank creation and movement of entities.

Figure 1 demonstrates a simple QA task where AMN constructs two memory banks based on the input. During inference only the entities in the left bank are considered reducing inference times. To realize its goals, AMN introduces a novel bank controller that uses reparameterization trick to make discrete decisions with high accuracy while maintaining differentiability. Finally, AMN also models sentence structures on-the-fly and propagates update information for all entities that allows it to solve all 20 bAbI tasks.

## 2 RELATED WORK

**Memory Networks:** Memory networks store the entire input sequence in memory and perform a softmax over hidden states to update the controller (Weston et al. (2014);Sukhbaatar et al. (2015)). DMN+ connects memory to input tokens and updates them sequentially (Xiong et al. (2016)). For inputs that consist of large number of tokens or entities, these methods can be expensive during inference. AMN stores entities with tied weights in different memory banks. By controlling the number of memory banks, AMN achieves low inference times with reasonable accuracy. Nearest neighbor methods have also been explored over memory networks. For example, Hierarchical Memory Networks separates the input memory into groups using the MIPS algorithm (Chandar et al. (2016)) . However, using MIPS is as slow as a softmax operation, so the authors propose using an approximate MIPS that gives inferior performance. In contrast, AMN is end to end differentiable, and reasons which entities are important and constructs a network with dynamic depth.

Neural Turing Machine (NTM) consists of a memory bank and a differentiable controller that learns to read and write to specific locations (Graves et al. (2014)). In contrast to NTMs, AMN memory bank controller is more coarse grained and the network learns to store entities in memory banks instead of specific locations. AMN uses a discrete bank controller that gives improved performance for bank controller actions over NTM's mechanisms. However, like NTMs, our design is consistent with the modeling studies of working memory by (Hazy et al. (2006)) where the brain performs robust memory maintenance and may maintain multiple working representations for individual working

tasks. Sparse access memory uses approximate nearest neighbors (ANN) to reduce memory usage in NTMs (Rae et al. (2016)). However, ANNs are not differentiable. AMN, uses a input specific memory organization that does not create sparse structures. This limits access during inference to specific entities reducing inference times.

Graph-based networks, (GG-NNs, Li et al. (2015) and GGT-NNs, Johnson (2017)) use nodes with tied weights that are updated based on gated-graph state updates with shared weights over edges. However, unlike AMN, they require strong supervision over the input and teacher forcing to learn the graph structure. Furthermore, the cost of building and training these models is expensive and if every edge is considered at every time-step the amount of computation grows at the order of $O(N^3)$ where $N$ represents the number of nodes/entities. AMN does not use strong supervision but can solve tasks that require transitive logic by modeling sentence walks on the fly. EntNet constructs dynamic networks based on entities with tied weights for each entity (Henaff et al. (2017)). A key-value update system allows it to update relevant (learned) entities. However, Entnet uses soft-attention during inference to attend to all entities that incur high inference costs. To summarize, majority of the past work on memory networks uses softmax over memory nodes, where each node may represent input or an entity. In contrast, AMN learns to organize memory into various memory banks and performs decode over fewer entities reducing inference times.

**Conditional Computation & Efficient Inference:** AMN is also related to the work on conditional computation which allows part of networks to be active during inference improving computational efficiency (Bengio et al. (2015)). Recently, this has been often accomplished using a gated mixture of experts (Eigen et al. (2013); Shazeer et al. (2017)). AMN conditionally attends to entities in initial banks during inference improving performance. For faster inference using CNNs, pruning (Le Cun et al. (1989); Han et al. (2016)), low rank approximations (Denton et al. (2014)), quantization and binarization (Rastegari et al. (2016)) and other tricks to improve GEMM performance (Vanhoucke et al. (2011)) have been explored. For sequence based inputs, pruning and compression has been explored (Giles & Omlin (1994); See et al. (2016)). However, compression results in irregular sparsity that reduces memory costs but may not reduce computation costs. Adaptive computation time (Graves (2016)) learns the number of steps required for inferring the output and this can also be used to reduce inference times (Figurnov et al. (2016)). AMN uses memory networks with dynamic number of banks to reduce computation costs.

**Dynamic networks:** Dynamic neural networks that change structure during inference have recently been possible due to newer frameworks such as Dynet and PyTorch. Existing work on pruning can be implemented using these frameworks to reduce inference times dynamically like dynamic deep network demonstrates (Liu & Deng (2017)). AMN utilizes the dynamic architecture abilities to construct an input dependent memory network of variable memory bank depth and the dynamic batching feature to process a variable number of entities. Furthermore, unlike past work that requires an apriori number of fixed memory slots, AMN constructs them on-the-fly based on the input. The learnable discrete decision-making process can be extended to other dynamic networks which often rely on REINFORCE to make such decisions (Liu & Deng (2017)).

**Neuroscience:** Our network construction is inspired by work on working memory representations. There is sufficient evidence for multiple, working memory representations in the human brain (Hazy et al. (2006)). Semantic memory (Tulving et al. (1972)), describes a hierarchical organization starting with relevant facts at the lowest level and progressively more complex and distant concepts at higher levels. AMN constructs entities from the input stories and stores the most relevant entities based on the question in the lowest level memory bank. Progressively higher level memory banks represent distant concepts (and not necessarily higher level concepts for AMN). Other work demonstrates organization of human memory in terms of "priority structure" where attention is a gate-keeper of working memory-guided by executive control's goals, plans, and intentions as in Watzl (2017), similar in spirit to AMN's question guided network construction.

## 3 DIFFERENTIABLE ADAPTIVE MEMORY MODULE

In this section, we describe the design process and motivation of our memory module. Our memory network architecture is created during inference time for every story. The architecture consists of

different memory banks and each memory bank stores entities from the input story. Hence, a *memory entity* represents the hidden state of each entity (each word in our case) from the input story while a *memory bank* is a collection of entities. Intuitively, each memory bank stores entities that have a similar distance score from the question.

At a high level, entities are gradually and recurrently copied through memory banks to *filter* out irrelevant nodes such that in the final inference stage, fewer entities are considered by the decoder. Note that the word *filter* implies a discrete decision and that recurrence implies time. If we were to perform a strict cut off and remove entities that appear to be irrelevant at each time step, learning the reasoning logic that requires previous entities that were cut off would not be possible. Thus, smoothed discretization is required.

We design filtering to be a two-stage pseudo-continuous process to simulate discrete cut offs ($\Pi_{move}, \Pi_{new}$), while keeping reference history. The overall memory ($M$) consists of multiple memory banks. A memory bank is a collection or group of entities ($m_{0...l}$), where $m_0$ denotes the initial and most general bank and $m_l$ denotes the most relevant bank. Note that $|l|$ is input dependent and learned. First, entities are moved from $m_0$ gradually towards $m_l$ based off of their individual relevance to the question and second, if $m_l$ becomes too saturated, $m_{l+1}$ is created. Operations in the external memory allowing for such dynamic restructuring and entity updates are described below. Note that these operations still maintain end to end differentiability.

1. Memory bank creation ($\Pi_{new}$), which creates a new memory bank depending on the current states of entities $m_i$. If the entropy, or information contained (explained below), of $m_i$ is too high, $\Pi_{new}(m_i)$ will learn to create a new memory bank $m_{i+1}$ to reduce entropy.

2. Moving entities across banks ($\Pi_{move}$), which determines which entities are relevant to the current question and move such entities to further (higher importance) memory banks.

3. Adding/Updating entities in a bank ($\Pi_{au}$), which adds entities that are not yet encountered to the first memory bank $m_0$ or if the entity is already in $m_0$, the operation updates the entity state.

4. Propagating changes across entities ($\Pi_{prop}$), which updates the entity states in memory banks based on node current states $\Pi_{prop}(M)$ and their semantic relationships. This is to communicate transitive logic.

Both $\Pi_{new}, \Pi_{move}$ require a discrete decision (refer to section 4.2.1.), and in particular, for $\Pi_{new}$ we introduce the notion of *entropy*. That is to say if $m_i$ contains too many nodes (the entropy becomes too high), the memory module will learn to create a new bank $m_{i+1}$ and move nodes to $m_{i+1}$ to reduce entropy. By creating more memory banks, the model spreads out the concentration of information which in turn better discretizes nodes according to relevance.

## 4 ADAPTIVE MEMORY NETWORKS

A high-level overview is shown in Figure 2, followed by a mathematical detail of the model's modules. Our model adopts the encoder-decoder framework with an augmented adaptive memory module. For an overview of the algorithm, refer to Section A.1.

**Notation and Problem Statement:** Given a story represented by $N$ **input** sentences (or statements), i.e., $(l_1, \cdots, l_N)$, and a **question** $q$, our goal is to generate an **answer** $a$. Each sentence $l$ is a sequence of $N$ words, denoted as $(w_1, \cdots, w_{N_s})$, and a question is a sequence of $N_q$ words denoted as $(w_1, \cdots, w_{N_q})$. Throughout the model we refer to entities; these can be interpreted as a 3-tuple of $\mathbf{e}_w = $ (word ID $wi$, hidden state $\mathbf{w}$, question relevance strength $\mathbf{s}$). Scalars, vectors, matrices, and dot products are denoted by lower-case letters, boldface lower-case letters and boldface capital letters, and angled brackets respectively.

### 4.1 ENCODER

The input to the model, starting with the encoder, are story-question input pairs. On a macro level, sentences $l_{1...N}$ are processed. On a micro level, words $w_{1...N_s}$ are processed within sentences.

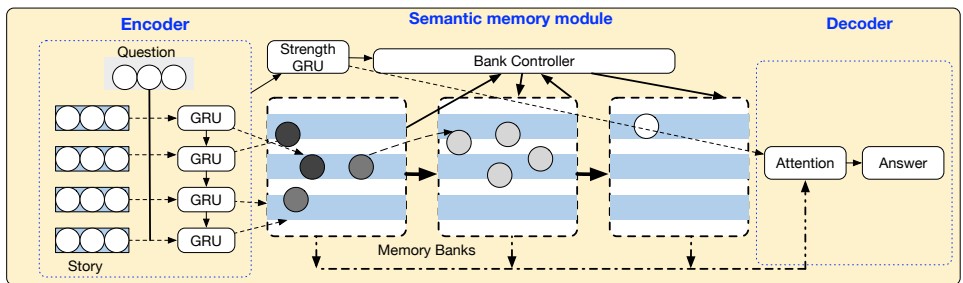

Figure 2: **Adaptive memory networks.**

For each $w_i \in l_i$, the encoder maps $w_i$ to a hidden representation and a question relevance strength $\in [0,1]$. The word ID of $w_i$ is passed through a standard embedding layer and then encoded through an accumulation GRU. The accumulation GRU captures the entity states through time by adding the output of each GRU time step to its respective word, stored in a lookup matrix. The initial states of $\mathbf{e}_w$ are set to this GRU output. Meanwhile, the question is also embedded and encoded in the same manner sans accumulation.

In the following, the subscripts $i, j$ are used to iterate through the total number of words in a statement and question respectively, $\mathbf{D}$ stores the accumulation GRU output, and $\mathbf{w}_i$ is a GRU encoding output. The last output of the GRU will be referred to as $\mathbf{w}_N, \mathbf{w}_{N_q}$ for statements and questions.

$$\mathbf{u}_i, \mathbf{u}_j = EMBED(wi_i), EMBED(wi_j) \quad (1) \qquad \mathbf{D}[i] \mathrel{+}= \mathbf{w}_i \quad (3)$$

$$\mathbf{w}_i = GRU(\mathbf{u}_i, \mathbf{w}_{i-1}) \quad (2) \qquad\qquad \mathbf{w}_j = GRU(\mathbf{u}_j, \mathbf{w}_{j-1}) \quad (4)$$

To compute the question relevance strength $s \in [0,1]$ for each word, the model uses GRU-like equations. The node strengths are first initialized to Xavier normal and the inputs are the current word states $\mathbf{w}^{in}$, the question state $\mathbf{w}_{N_q}$, and when applicable, the previous strength. Sentences are processed each time step $t$.

$$\mathbf{z}^t = \sigma(\mathbf{U}_z \mathbf{w}^{in} + \mathbf{W}_z \mathbf{w}_{N_q} + \mathbf{X}_z \mathbf{s}^{t-1}) \quad (5) \qquad \widetilde{\mathbf{s}}^t = \sigma(\mathbf{W}_h \mathbf{w}^{in} + \mathbf{U}_h(\mathbf{r}^t \odot \mathbf{s}^{t-1})) \quad (7)$$

$$\mathbf{r}^t = 1 - \sigma(\mathbf{U}_r \langle \mathbf{s}^{t-1}, \mathbf{w}_{N_q} \rangle) \quad (6) \qquad\qquad \mathbf{s}^t = \mathbf{z}^t \odot \mathbf{s}^{t-1} + (1 - \mathbf{z}^t) \odot \widetilde{\mathbf{s}}^t \quad (8)$$

In particular, equation (6) shows where the model learns to lower the strengths of nodes that are not related the question. First, a dot product between the current word states and question state are computed for similarity (high correlation), then it is subtracted from a 1 to obtain the dissimilarity. We refer to these operations as SGRU (Strength GRU) in Algorithm 1.

## 4.2 ADAPTIVE MEMORY MODULE

The adaptive memory module recurrently restructures entities in a question relevant manner so the decoder can then consider fewer entities (namely, the question relevant entities) to generate an answer. The following operations are performed once per sentence.

### 4.2.1 MEMORY BANK CONTROLLER

As mentioned earlier, discrete decisions are difficult for neural networks to learn so we designed a specific memory bank controller $\Pi_{ctrl}$ for binary decision making. The model takes ideas from the reparameterization trick and uses custom backpropagation to maintain differentiability.

In particular, the adaptive memory module needs to make two discrete decisions on a $\{0,1\}$ basis, one in $\Pi_{new}$ to create a new memory bank and the other in $\Pi_{move}$ to move nodes to a different memory bank. The model uses a scalar $p \in \{0,1\}$ to parameterize a Bernoulli distribution where the realization $\mathbf{H}$, is the decision the model makes. However, backpropagation through a random node is intractable, so the model detaches $\mathbf{H}$ from the computation graph and introduces $\mathbf{H}$ as a new node. Finally, $\mathbf{H}$ is used as a mask to zero out entities in the discrete decision. Meanwhile, $p$ is kept in the computation graph and has a special computed loss (Section 4.4). The operations below will

be denoted as $\Pi_{ctrl}$ and has two instances: one for memory bank creation $\Pi_{new}$ and one for moving entities across banks $\Pi_{move}$. In equation 9, depending on what $\Pi_{ctrl}$ is used for, $q$ is a polymorphic function and will take on a different operation and $*$ will be a different input. Examples of such are given in their respective sections (4.2.2.1, 4.2.2.2).

$$p = q(*) \qquad (9) \qquad\qquad \mathbf{H} = \text{Bernoulli}(p) \qquad (10)$$

### 4.2.2 Memory Bank Operations

1. **Memory bank creation $\Pi_{new}$:** To determine when a new memory bank is created, in other words, if the current memory bank becomes too saturated, the memory bank controller (4.2.1.) will make a discrete decision to create a new memory bank. Here, $q$ (eq 9) is a fully connected layer and the input is the concatenation of all the current memory bank $m_i$'s entity states $[\mathbf{w}_0...\mathbf{w}_i] \in R^{1,n|\mathbf{e}_w|}$. Intuitively, $q$ will learn a continuous decision that is later discretized by eq 10 based on entity states and the number of entities. Note this is only performed for the last memory bank.

$$\Pi_{new}([\mathbf{w}_0...\mathbf{w}_i]) = \begin{cases} \mathbf{M}.\text{new}() & \text{if } 1(\Pi_{ctrl}([\mathbf{w}_0...\mathbf{w}_i])) \text{ else} \\ \text{pass} \end{cases} \qquad (11)$$

2. **Moving entities through memory banks:** Similar to $\Pi_{new}$, individual entities' relevance scores are passed into the bank controller to determine $\mathbf{H}$ as the input. The relevance score is computed by multiplying an entity state by its respective relevance $\in R^{n,|\mathbf{e}_w|}$. Here, $q$ has a slight modification and is the identity function. Note that this operation can only be performed if there is a memory bank to move nodes to, namely if $m_{i+1}$ exists. Additionally, each bank has a set property where it cannot contain duplicate nodes, but the same node can exist in two different memory banks.

$$\Pi_{move}(\mathbf{s}_i * \mathbf{w}_i) = m.\text{add}(1(\Pi_{ctrl}(\mathbf{s}_i * \mathbf{w}_i))) \quad \forall i \in m \qquad (12)$$

3. **Adding/Updating entities in a bank:** Recall that entities are initially set to the output of $\mathbf{D}$. However, as additional sentences are processed, new entities and their hidden states are observed. In the case of a new entity $\mathbf{e}_w$, the entity is added to the first memory bank $m_0$. If the entity already exists in $m_0$, then $\mathbf{e}_w$'s corresponding hidden state is updated through a GRU. This procedure is done for all memory banks.

$$\Pi_{au}([\mathbf{w}_0...\mathbf{w}_i]) = \begin{cases} m_0.\text{add}(\mathbf{e}_w^i) & \text{if } \mathbf{e}_w^i \notin m_0 \text{ else} \\ \mathbf{w}_i^{t+1} = GRU(\mathbf{w}_N, \mathbf{w}_i^t) & \forall m \in M \end{cases} \qquad (13)$$

4. **Propagating updates to related entities:** So far, entities exist as a bag of words model and the sentence structure is not maintained. This can make it difficult to solve tasks that require transitive reasoning over multiple entities. To track sentence structure information, we model semantic relationships as a directed graph stored in adjacency matrix $\mathbf{A}$. As sentences are processed word by word, a directed graph is drawn progressively from $w_0...w_i...w_N$. If sentence $l_k$'s path contains nodes already in the current directed graph, $l_k$ will include said nodes in its path. After $l_k$ is added to $\mathbf{A}$, the model propagates the new update hidden state information $\mathbf{a}_i$ among all node states using a GRU. $\mathbf{a}_i$ for each node $i$ is equal to the sum of the incoming edges' node hidden states.

Additionally, we add a particular emphasis on $l_k$ to simulate recency. At face value, one propagation step of $\mathbf{A}$ will only have a reachability of its immediate neighbor, so to reach all nodes, $\mathbf{A}$ is raised to a consecutive power $r$ to reach and update each intermediate node. $r$ can be either the longest path in $\mathbf{A}$ or a set parameter. Again, this is done within a memory bank for all memory banks. For entities that have migrated to another bank, the update for these entities is a no-op but propagation information as per the sentence structure is maintained. A single iteration is shown below:

$$\mathbf{a} = (\mathbf{A}^r)^T[\mathbf{w}_0...\mathbf{w}_i] \quad (14) \qquad\qquad \mathbf{w}^t = GRU(\mathbf{a}, \mathbf{w}^{t-1}) \quad (15)$$

When nodes are transferred across banks, $\mathbf{A}$ is still preserved. If intermediate nodes are removed from a path, a transitive closure is drawn if possible.

After these steps are finished at the end of a sentence, namely, the memory unit has reasoned through how large (number of memory banks) the memory should be and which entities are relevant at the current point in the story, all entities are passed through the strength modified GRU (4.1, eq 5-8) to recompute their question relevance (relevance score).

## 4.3 DECODE

After all sentences $l_{1...N}$ are ingested, the decode portion of the network learns to interpret the results from the memory banks. The network iterates through the memory banks using a standard attention mechanism. To force the network to understand the question importance weighting, the model uses an exponential function $d$ to weight important memory banks higher. $\mathbf{C}_m$ are the hidden states contained in memory $m$, $\mathbf{s}_m$ are the relevance strengths of memory bank $m$, $\mathbf{w}_{N_q}$ is the question hidden state, $\mathbf{ps}$ is the attention score, $r, h$ are learned weight masks, $\mathbf{g}$ are the accumulated states, and $\mathbf{l}$ is the final logits prediction. During *inference*, fewer memory banks are considered.

$$\mathbf{C}_m = \mathbf{s}_m \cdot [\mathbf{w}_0, ...\mathbf{w}_i] \quad \forall i \in m \quad (16) \qquad \mathbf{g} \mathrel{+}= d(\langle \mathbf{C}_m, \mathbf{ps}\rangle) \quad (18)$$

$$\mathbf{ps} = Softmax(\langle \mathbf{C}_m, \mathbf{w}_{N_q}\rangle) \quad (17) \qquad \hat{\mathbf{L}} = r(\text{PReLU}(h(\mathbf{g}) + \mathbf{w}_{N_q}) \quad \text{if } m \text{ is last} \quad (19)$$

## 4.4 LOSS

Loss is comprised of two parts, answer loss, which is computed from the given annotations, and secondary loss (from $\Pi_{new}$, $\Pi_{move}$), which is computed from sentence and story features at each sentence time step $l_{0...N}$. Answer loss is standard cross entropy at the end of the story after $l_N$ is processed.

$$\mathcal{L}_p(\hat{\mathbf{L}}) = \text{CrossEntropy}(\hat{\mathbf{L}}, \mathbf{L})$$

After each sentence $l_i$, the node relevance $\mathbf{s}_{l_i}$ is enforced by computing the expected relevance $\mathbb{E}[\mathbf{s}_{l_i}]$. $\mathbb{E}[\mathbf{s}]$ is determined by nodes that are connected to the answer node $\mathbf{a}$ in a directed graph; words that are connected to $\mathbf{a}$ are relevant to $\mathbf{a}$. They are then weighted with a deterministic function of distance from $\mathbf{a}$.

$$\mathcal{L}_s(\mathbf{s}) = D_{KL}(\mathbf{s}_{l_i}||\mathbb{E}[\mathbf{s}_{l_i}])$$

Additionally, bank creation is kept in check by constraining $p_{l_i}$ w.r.t. the expected number of memory banks. The expected number of memory banks can be thought of as a geometric distribution $\sim \text{Geometric}(\hat{p}_{l_i})$ parameterized by $\hat{p}_{l_i}$, a hyperparameter. Typically, at each sentence step $\hat{p}$ is raised to the inverse power of the current sentence step to reflect the amount of information ingested. Intuitively, this loss ensures banks are created when a memory bank contains too many nodes. On the other hand, the learned mask $q$ (eq. 9) enables the model to weight certain nodes a higher entropy to prompt bank creation. Through these two dependencies, the model is able to simulate bank creation as a function of the number of nodes and the type of nodes in a given memory bank.

$$\mathcal{L}_b(p_{l_i}) = D_{KL}(p_{l_i}||\hat{p}^{\frac{1}{\beta|l_i|}})$$

All components combined, the final loss is given in the following equation

$$\mathcal{L}_{total} = \mathcal{L}_p(\hat{\mathbf{L}}) + \sum_{i=1}^{|l_n|}(\mathcal{L}_s^i(\mathbf{s}) + \mathcal{L}_b^i(p))$$

## 5 EVALUATION

In this section, we evaluate AMN accuracy and inference times on the bAbI dataset Weston et al. (2015) and extended bAbI tasks dataset. We compare our performance with Entnet (Henaff et al.

(2017)), which recently achieved state of the art results on the bAbi dataset. For accuracy measurements, we also compare with DMN+ and encoder-decoder methods. Finally we discuss the time trade offs between AMN and current SOTA methods. The portion regarding inference times are not inclusive of story ingestion. We summarize our experiments results as follows:

- We are able to solve all bAbi tasks using AMN. Furthermore, AMN is able to reason important entities and propagate them to the final memory bank allowing for 48% fewer entities examined during inference.

- We construct extended bAbI tasks to evaluate AMN behavior. First, we extend Task 1 for *multiple questions* in order to gauge performance in a more robust manner. For example, if a reasonable set of questions are asked (where reasonable means that collectively they do not require all entities to answer implying entities can be filtered out), will the model still sufficiently reason through entities. We find that our network is able to reason useful entities for both tasks and store them in the final memory bank. Furthermore, we also scale bAbI for a large number of entities and find that AMN provides additional benefits at scale since only relevant entities are stored in the final memory bank.

## 5.1 EXPERIMENT SETTINGS

We implement our network in PyTorch (Paszke et al. (2017)). We initialize our model using Xavier initialization, and the word embeddings utilize random uniform initialization ranging from $-\sqrt{3}$ to $\sqrt{3}$. The learning rate is set as $0.001$ initially and updated with a learning rate scheduler. $\mathbb{E}[\mathbf{s}]$ contains nodes in the connected components of $\mathbf{A}$ containing the answer node $\mathbf{a}$ which has relevance scores sampled from a Gaussian distribution centered at $0.75$ with a variance of $0.05$ (capped at 1). Nodes that are not in the connected component containing $\mathbf{a}$ are similarly sampled from a Gaussian centered from $0.3$ with a variance of $0.1$ (capped at 0). $\hat{p}_{l_i}$ is initially set to $0.8$ and $\beta$ varies depending on the story length from $0.1 \leq \beta \leq 0.25$. Note that for transitive tasks, $\hat{p}_{l_i}$ is set to $0.2$. We train our models using the Adam optimizer (Kingma & Ba, 2014).

## 5.2 BABI DATASET

The bAbI task suite consists of 20 reasoning tasks that include deduction, induction, path finding etc. Results are from the following parameters: $\leq 200$ epochs, best of 10 runs. Table 1 shows the accuracy and Table 4 shows the inference performance in terms of the number of entities examined. A task is considered passed if the error rate is less than 5%.

We find that AMN creates $1-6$ memory banks for different tasks. We also find that 8 tasks can be solved by looking at just one memory bank and 14 tasks can be solved with half the total number of memory banks. Lastly, all tasks can be solved by examining less than or equal the total number of entities ($e \in M \leq |V| + \epsilon$)[1]. Tasks that cannot be solved in fewer than half the memory banks either require additional entities due to transitive logic or have multiple questions. For transitive logic, additional banks could be required as an relevant nodes may be in a further bank. However, this still avoids scanning all banks. In the case of multiple questions, all nodes may become necessary to construct all answers. We provide additional evaluation in Appendix to examine memory bank behavior for certain tasks.

**Inference performance** Table 4 shows the number of banks created and required to solve a task, as well as the ratio of entities examined to solve the task. Table 3 shows the complexity of AMN and other SOTA models. Entnet uses an empirically selected parameter, typically set to the number of vocabulary words. GGT-NN uses the number of vocabulary words and creates new $k$ new nodes intermittently per sentence step.

For tasks where nodes are easily separable where nodes are clearly irrelevant to the question(s), AMN is able to successfully reduce the number of nodes examined. However for tasks that require more information, such as counting (Task 7), the model is still able to obtain the correct answer

---

[1] The entities used to construct an answer *and pass the task* are examined as the sum of all entities across the $M$ which is usually $O(|V|)$. However, this is within an error margin of 6% more entities on some experiments, and thus we included an $\epsilon$ term.

| Task | AMN | Entnet | DMN+ | MemN2N | EncDec |
|------|-----|--------|------|--------|--------|
| 1 - Single Supporting Fact | **0.0** | 0.0 | 0.0 | 0.0 | 52.0 |
| 2 - Two Supporting Facts | **2.1** | 0.1 | 0.3 | 0.3 | 66.1 |
| 3 - Three Supporting Facts | 4.7 | 4.1 | 1.1 | 2.1 | 71.9 |
| 4 - Two Arg. Relations | **0.0** | 0.0 | 0.0 | 0.0 | 29.2 |
| 5 - Three Arg. Relations | **2.7** | 0.3 | 0.5 | 0.8 | 14.3 |
| 6 - Yes/No Questions | 3.1 | 0.2 | 0.0 | 0.1 | 31.0 |
| 7 - Counting | 0.0 | 0.0 | 2.4 | 2.0 | 21.8 |
| 8 - Lists/Sets | 0.0 | 0.5 | 0.0 | 0.9 | 27.6 |
| 9 - Simple Negation | 1.3 | 0.1 | 0.0 | 0.3 | 36.4 |
| 10 - Indefinite Knowledge | **1.2** | 0.6 | 0.0 | 0.0 | 36.4 |
| 11 - Basic Coreference | **2.7** | 0.3 | 0.0 | 0.1 | 31.7 |
| 12 - Conjunction | **2.2** | 0.0 | 0.0 | 0.0 | 35.0 |
| 13 - Compound Coref. | 4.6 | 1.3 | 0.0 | 0.0 | 6.80 |
| 14 - Time Reasoning | **2.1** | 0.0 | 0.2 | 0.1 | 67.2 |
| 15 - Basic Deduction | **1.8** | 0.0 | 0.0 | 0.0 | 62.2 |
| 16 - Basic Induction | 4.2 | 0.0 | 45.3 | 51.8 | 54.0 |
| 17 - Positional Reasoning | **4.3** | 0.5 | 4.2 | 18.6 | 43.1 |
| 18 - Size Reasoning | 2.0 | 0.3 | 2.1 | 5.3 | 6.60 |
| 19 - Path Finding | 2.4 | 2.3 | 0.0 | 2.3 | 89.6 |
| 20 - Agents Motivations | **0.0** | 0.0 | 0.0 | 0.0 | 2.30 |
| No. of failed tasks (>5%) | **0** | **0** | 5 | 6 | 20 |

Table 1: Performance comparison of various models in terms of test error rate (%) and the number of failed tasks on the bAbI dataset. The bold task scores are where AMN can solve the task using only 1 memory bank.

| Task | Created Banks (Rounded Average) | Required Banks | Ratio ($\frac{e \in M}{|V|}$) |
|------|----------------------------------|----------------|-------------------------------|
| 1 - Single Supporting Fact | 3 | 1 | 0.22 |
| 2 - Two Supporting Facts | 5 | 1 | 0.41 |
| 4 - Two Arg. Relations | 2 | 1 | 0.70 |
| 7 - Counting | 5 | 2 | 0.81 |
| 10 - Indefinite Knowledge | 1 | 1 | 1.00 |
| 11 - Basic Coreference | 3 | 1 | 0.43 |
| 12 - Conjunction | 2 | 1 | 0.37 |
| 14 - Time Reasoning | 3 | 1 | 0.60 |
| 15 - Basic Deduction | 1 | 1 | 1.00 |
| 16 - Basic Induction | 2 | 2 | *1.06* |
| 17 - Positional Reasoning | 1 | 1 | 1.00 |
| 18 - Size Reasoning | 3 | 2 | 0.82 |
| 19 - Path Finding | 2 | 2 | *1.05* |
| 20 - Agents Motivations | 2 | 1 | 0.26 |
| Extended bAbi | | | |
| 1 - Single Supporting Fact, 100 Entities | 6 | 1 | .13 |
| 1 - Single Supporting, Multiple Questions | 3 | 1 | .38 |

Table 2: Memory bank analysis of indicative tasks.

without using all entities. Lastly, transitive logic tasks where information is difficult to separate due to dependencies of entities, the model creates very few banks (1 or 2) and uses all nodes to correctly generate an answer. We note that in the instance where the model only creates one bank, it is very sparse, containing only one or two entities.

Because variations in computation times in text are minute, the number of entities required to construct an answer are of more interest as they directly correspond to the number of computations required. Additionally, due to various implementations of current models, their run times can significantly vary. However, for the comparison of inference times, AMN's decoder and EntNet's decoder are highly similar and contain roughly the same number of operations.

## 5.3 EXTENDED BABI TASKS

We extend the bAbI tasks by adding additional entities and sentences and adding multiple questions for a single story, for Task 1.

**Scaled Task 1:** We increase the the number of entities to 100 entities in the task generation system instead of existing 38. We also extend the story length to 90 to ensure new entities are referenced. We find that AMN creates 6 memory banks and the ratio of entities in the final banks versus the overall entities drops to 0.13 given the excess entities that are not referenced in the questions.

**Multiple questions:** We also augment the tasks with multiple questions to understand if AMN can handle when a story has multiple questions associated with it. We extend our model to handle multiple questions at once to limit re-generating the network for every question. To do so, we modify bAbi to generate several questions per story for tasks that do not currently have multiple questions. For single supporting fact (Task 1), the model creates 3 banks and requires 1 bank to successfully pass the task. *Furthermore*, the ratio of entities required to pass the task only increases by 0.16 for a total of 0.38.

## 6 CONCLUSION AND FUTURE WORK

In this paper, we present Adaptive Memory Network that learns to adaptively organize the memory to answer questions with lower inference times. Unlike NTMs which learn to read and write at individual memory locations, Adaptive Memory Network demonstrates a novel design where the learned memory management is coarse-grained that is easier to train.Through our experiments, we demonstrate that AMN can learn to reason, construct, and sort memory banks based on relevance over the question set.

AMN architecture is generic and can be extended to other types of tasks where the input sequence can be separated into different entities. In the future, we plan to evaluate AMN over such tasks to evaluate AMN generality. We also plan to experiment with larger scale datasets (beyond bAbI, such as a document with question pairs) that have a large number of entities to further explore scalability.

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

| Method | Complexity |
|---|---|
| Entnet (Henaff et al. (2017)) | $O(|V|)$ |
| GGT-NN (Johnson (2017)) | $O(|V| + kS)$ |
| **AMN (ours)** | $O(\alpha|V|)) : 0 < \alpha < 1 + \epsilon$ |

Table 3: Comparison of decode complexity for AMN, Entnet and GGT-NN.

# A  APPENDIX

## A.1  ALGORITHM

We describe our overall algorithm in pseudo-code in this section. We follow the notation as described in the paper.

---
**Algorithm 1** AMN($\mathbf{S}, \mathbf{q}, \mathbf{a}$)
---

1: $\mathbf{M} \leftarrow \varnothing$
2: **for** sentence $s \in \mathbf{S}$ **do**
3:     **for** word $w \in$ s **do**
4:       $\mathbf{D} \leftarrow \text{ENCODE}(w, \mathbf{q})$
5:     **end for**
6:     $\mathbf{n}_{m_i} \leftarrow \text{SGRU}(\mathbf{D})$
7:     **for** memory bank $m_i \in \mathbf{M}$ **do**
8:       $m_i \leftarrow \Pi_{au}(m_i, \mathbf{D})$
9:       $m_i \leftarrow \Pi_{prop}(m_i)$
10:       $m_{i+1} \leftarrow \Pi_{move}(m_i, \mathbf{n}_{m_i})$
11:       $\mathbf{n}_{m_i} \leftarrow \text{SGRU}(\mathbf{D}, \mathbf{n}_{m_i})$
12:       **if** $i = |\mathbf{M}|$ and $\Pi_{new}(m_i)$ **then**
13:         $\mathbf{M}, p \leftarrow [\mathbf{M}, m_{i+1}]$
14:         Repeat 8 to 11 once
15:       **end if**
16:     **end for**
17: **end for**
18: $\hat{\mathbf{a}} \leftarrow \text{DECODE}(\mathbf{M}, \mathbf{q})$

---

## A.2  DECODE OVERHEAD

We compare the computations costs during the decode operation during inference for solving the extended bAbi task. We compute the overheads for AMN Entnet (Henaff et al. (2017)) and GGT-NN. Table 3 gives the decode comparisons between AMN, Entnet and GGT-NN. Here, $|V|$ represents to the total number of entities for all networks. GGT-NN can dynamically create nodes and $k$ k is hyper parameter the new nodes created for $S$ sentences in input story. $\alpha$ is the percent of entities stored in the final bank w.r.t to the total entities for AMN.

We compare the wall clock execution times for three tasks within bAbI for 1000 examples/task. We compare the wall-clock times for three tasks. We compare the inference times of considering all banks (and entities) versus the just looking at the passing banks as required by AMN. We find that AMN requires fewer banks and as a consequence fewer entities and saves inference times.

| Task | Created Banks | Required Banks | Baseline (All banks) | AMN (Required banks) |
|---|---|---|---|---|
| 1 - Single Supporting Fact | 3 | 1 | 2.15 s | 0.6 s |
| 2 - Two Supporting Facts | 5 | 1 | 15.8 s | 3.2 s |
| 7 - Counting | 5 | 2 | 21 s | 6.0 s |

Table 4: Memory bank wall clock times for representative tasks for 1000 examples (time in secs).

## A.3  MEMORY BANK BEHAVIOR

In this section, we understand memory bank behavior of AMN. Figure 3 shows the memory banks and the entity creation for a single story example, for some of the tasks from bAbI. Depending upon the task, and distance from the question AMN creates variable number of memory banks. The heatmap demonstrates how entities are copied across memory banks. Grey blocks indicate absence of those banks.

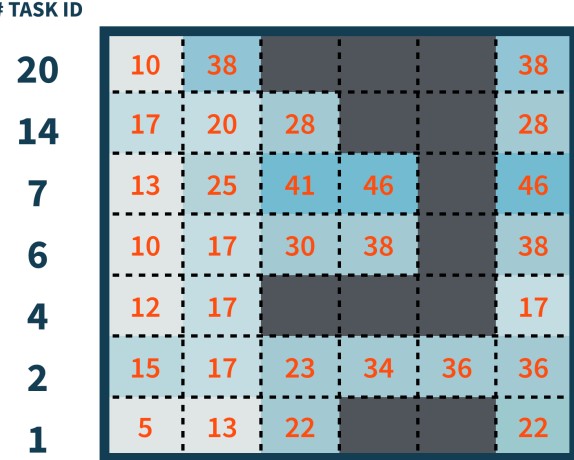

Figure 3: Heat map showing distribution of entities across various memory banks for simple bAbI tasks. The x-axis shows the task IDs (refer to Table 1 for task details for each ID.)

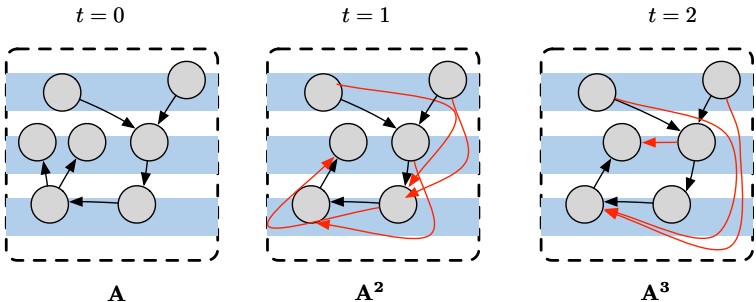

Figure 4: Propagation in AMN (shown for a single memory bank across time).

### A.4   PROPAGATION EXAMPLE

In this section, we explain propagation with an example. Figure 4 shows how propagation happens after every time step. The nodes represent entities corresponding to words in a sentence. As sentences are processed word by word, a directed graph is drawn progressively from $w_0...w_i...w_N$. If sentence $l_k$'s path contains nodes already in the current directed graph, $l_k$ will include said nodes in the its path. After $l_k$ is added to $\mathbf{A}$, the model propagates the new update hidden state information $\mathbf{a}_i$ among all node states using a GRU. $\mathbf{a}_i$ for each node $i$ is equal to the sum of the incoming edges' node hidden states. Additionally, we add a particular emphasis on $l_k$ to simulate recency. At face value, one propagation step of $\mathbf{A}$ will only have a reachability of its immediate neighbor, so to reach all nodes, $\mathbf{A}$ is raised to a consecutive power $r$ to reach and update each intermediate node. $r$ can be either the longest path in $\mathbf{A}$ or a set parameter.

