# OpenReview forum: "Adaptive Memory Networks"
_ICLR.cc/2018/Conference — Invite to Workshop Track_

### Official Review · AnonReviewer1 · 2017-11-27
**"faster inference" is not convincing.**

**Rating:** 7
**Confidence:** 5

**Review:**

Summary:

This paper proposes a dynamic memory augmented neural network for question answering. The proposed model iteratively creates a shorter list of relevant entities such that the decoder can look at only a smaller set of entities to answer the given question. Authors show results in bAbi dataset.

My comments:

1. While the proposed model is very interesting, I disagree with the claim that AMN has lower inference times. The memory creation happens only after reading the question and hence the entire process can be considered as part of inference. So it is not clear if there is a huge reduction in the inference time when compared to other models that the authors compare. However, the proposed model looks like a nice piece of interpretable reasoning module. In that sense, it is not any better than EntNet based on the error rate since EntNet is doing better than AMN in 15 out of 20 tasks. So it is not very clear what is the advantage of AMN over EntNet or other MANN architectures.

2. Can you explain equation 9 in detail? What is the input to the softmax function? What is the output size of the softmax? I assume q produces a scalar output. But what is the input size to the q function?

3. In the experiment, when you say “best of 10 runs”, is it based on a separate validation set? Please report the mean and variance of the 10 runs. It is sad that people just report best of multiple runs in the bAbi tasks and not report the variance in the performance. I would like to see the mean and variance in the performance.

4. What happens when number of entities is large? Can you comment about how this model will be useful in situations other than reading comprehension style QA?

5. Are the authors willing to release the code for reproducing the results?

Minor comments:

1. Page 2, second line: “Networks(AMN)” should be “Networks (AMN).
2. In page 3, first line: “Hazy et al. (2006)” should be “(Hazy et al. 2006)”.
3. In page 3, second para, first line, both references should use \citep instead of \citet.
4. In page 4, fourth para, Vanhoucke et al should also be inside \citep.
5. In page 4, notations paragraph: “a question is a sequence of N_q words” - “of” is missing.
6. In page 5, first paragraph is not clear.
7. In page 6, point 4, 7th line: “nodes in the its path” should be “nodes in its path”.
8. In page 9, section 5.3, multiple questions, 2nd line: “We extend the our model” should be “We extend our model”.

---

> ### Author Response · Authors · 2017-12-28
> **Thanks for your comments**
>
> We thank the reviewer for providing us with a detailed feedback.
>
> 1) Since AMN (our model) reduces the number of entities under test, the inference times is reduced. We have updated the draft with the inference time information in Appendix. This can be useful when questions (or hints) are available during the QA process (we describe the Amazon example in the Introduction). Your concern about memory creation on a per question basis in a memory network is a valid one. Therefore, we extend AMN to multiple questions as shown in the evaluation. Hence, given a list of questions, AMN can learn to construct a network architecture such that these questions can be answered quickly. Here, network construction costs are amortized for mutiple questions.
>
> 2) In equation 9, depending on what ΠCtrl is used for, Q is a polymorphic function and will take on a different operation and ∗ will be a different input. Examples of such are given in the paper in the respective sections (4.2.2.1, 4.2.2.2) with the required details.
>
> 3) The result is on the validation set. We share the frustration. However, this is how a majority of the past work is reported. We plan on providing variance and mean for Entnet, GGT-NN and our work in the final version of the paper.
>
> 4) When the number of entities is large, inference slows down. It also reduces the accuracy in some cases since the final operation such as softmax is performed over a large number of entities and it can be difficult to train. Our model can be applied to other QA tasks such as VQA. Here, each entity can additionally include CNN feature output and inference costs can be reduced.
>
> 5) Yes, we plan to release the code with the final version of the paper.
>
> We have fixed all the minor comments as you mention in your review. We really appreciate your help in improving the paper.

---

> > ### Comment · AnonReviewer1 · 2018-01-12
> > **Final note**
> >
> > I read the response from the authors. I was expecting to see the mean and variance of the 10 runs. Other responses are convincing. I still stand by my decision.
> >
> > If the paper gets accepted, please do report the mean and variance of your experiments.

---

> > > ### Author Response · Authors · 2018-01-17
> > > **Thanks**
> > >
> > > Thanks for taking time to revisit the paper. We are working on additional experiments and will add these results in the coming revision.

---

### Official Review · AnonReviewer2 · 2017-11-28

**Rating:** 4
**Confidence:** 3

**Review:**

The authors propose a model for QA that given a question and a story adaptively determines the number of  entity groups (banks). The paper is rather hard to follow as many task specific terms are not explained. For instance, it would benefit the paper if the authors introduced the definitions of a bank and a story. This will help the reader have a more comprehensive understanding of their framework.

The paper capitalized on the argument of faster inference and no wall-time for inference is shown. The authors only report the number of used banks. What are the runtime gains compared to Entnet?
This was the core motivation behind this work and the authors fail to discuss this completely.

---

> ### Author Response · Authors · 2017-12-25
> **thanks for your comments in improving the paper.**
>
> Thank you for your review. We have updated Section 3 and 4 to improve the readability of the paper. We also added the bank and entity definitions at the beginning of Section 3. Furthermore, we have uploaded a revision that provides the wall clock savings in Appendix A.3 for three tasks. As our results show, the wall clock times are directly proportional to the number of entities under consideration during inference. Adaptive Memory Network (AMN) architecture reduces inferences times by learning to attend to fewer entities during inference. Please let us know if you have revised comments based on the new draft.

---

### Official Review · AnonReviewer4 · 2017-11-30
**Very promising approach, but it seems the authors were not able to submit a finished manuscript on time.**

**Rating:** 5
**Confidence:** 4

**Review:**

This paper offers a very promising approach to the processing of the type of sequences we find in dialogues, somewhat in between RNNs which have problem modeling memory, and memory networks whose explicit modeling of the memory is too rigid.

To achieve that, the starting point seems to be a strength GRU that has the ability to dynamically add memory banks to the original dialogue and question sentence representations, thanks to the use of imperative DNN programming. The use of the reparametrization trick to enable global differentiability is reminiscent of an ICLR'17 paper "Learning graphical state transitions". Compared to the latter, the current paper seems to offer a more tractable architecture and optimization problem that does not require strong supervision and should be much faster to train.

Unfortunately, this is the best understanding I got from this paper, as it seems to be in such a preliminary stage that the exact operations of the SGRU are not parsable. Maybe the authors have been taken off guard by the new review process where one can no longer improve the manuscript during this 2017 review (something that had enabled a few paper to pass the 2016 review).

After a nice introduction, everything seems to fall apart in section 4, as if the authors did not have time to finish their write-up.
- N is both the number of sentences and number of word per sentence, which does not make sense.
- i iterates over both the sentences and the words.

The critical SGRU algorithm is impossible to parse
- The hidden vector sigma, which is usually noted h in the GRU notation, is not even defined
- The critical reset gate operation in Eq.(6) is not even explained, and modified in a way I do not understand compared to standard GRU.
- What is t? From algorithm 1 in Appendix A, it seems to correspond to looping over both sentences and words.
- The most novel and critical operation of this SGRU, to process the entities of the memory bank, is not even explained. All we get at the end of section 4.2 is " After these steps are finished, all entities are passed through the strength modified GRU (4.1) to recompute question relevance."

The algorithm in Appendix A does  not help much.  With PyTorch being so readable, I wish some source code had been made available.

Experiments reporting also contains unacceptable omissions and errors:
- The definition of 'failed task', essential for understanding, is not stated (more than 5% error)
- Reported numbers of failed tasks are erroneous: it should be 1 for DMN+ and 3 for MemN2N.

The reviewers corrections, while significant, do not seem enough to clarify the core of the paper.

Page 3: dynanet -> dynet

---

> ### Author Response · Authors · 2017-12-28
> **Thanks for your review**
>
> We are sorry about the difficulty in understanding our paper. We have posted a revision that fixes these concerns. Additionally, we have fixed and clarified the notation as necessary.
>
> The strength GRU measures the relevance of each word from the question. The equations in the paper now accompany additional, helpful text and the update is performed at a sentence level. We also fixed the algorithm in the appendix. The relevance score coupled with the memory bank design allow AMN to look at only relevant entities during inference time.
>
> We also fixed the tasks error rates and passing tasks definition. We have improved the readability of the paper. However, we plan to provide documented source code with the final version of the paper. We appreciate your comments in improving the paper. Please let us know if you have additional comments about the paper.

---

> ### Author Response · Authors · 2018-01-13
> **Available**
>
> Dear reviewer,
>
> The revised paper is available. We can see 03 Nov 2017 (modified: 05 Jan 2018) as the latest revision. Furthermore, if you click on revisions, you can see the diff between the latest draft and the submitted version by clicking on 'Compare Revisions' on the top right to see the changes we have made. Please let us know if you have additional concerns.
>
> Update 01/17: Thanks for your feedback and taking time to revisit the paper. We will work towards improving the paper.

---

### Public Comment · (anonymous) · 2017-11-21
**Confusion memory bank / memory slot**

The authors may wish to consult the definition of "bank" which is used extensively throughout the paper (as "memory bank") without a definition. A bank is supposed to be "series of similar things". It seems they use "memory bank" for "memory slot" making the text somewhat confusing.

---

> ### Author Response · Authors · 2017-11-22
> **bank consists of multiple entities**
>
> Thanks for reading our paper. We agree that the abstract does not describe memory bank/slot clearly. We plan to clarify the this in the next revision. A bank in the paper does mean a series of similar entities.
>
> An entity is a 3-tuple of word ID, hidden state, and a question relevance strength. A memory slot can hold this entity. A bank consists of multiple entities. The network learns to store various entities as the story is ingested into a bank. As the story is read in, the network learns to create newer banks and copy the entities. During inference, only a single bank or a few banks are used to answer the question, saving inference times.

---

### Author Response · Authors · 2017-12-31
**thanks for your comments**

We would like to thank all reviewers for all the comments and feedback towards improving the paper. We have fixed the typos, terms, and explanations in the paper and made it more accessible. We have also added inference times for representative tasks that are consistent with the savings in terms of the number of entities accessed during inference in Appendix A.2.

Adaptive Memory Networks (AMN) presents a dynamic network design where entities from the input stories are stored in memory banks. Starting from a single bank, as the number of input entities increases, the network learns to create new banks as the entropy in a single bank becomes too high. Over a period of time, the network represents a hierarchical structure where entities are stored in different banks distanced by the question. During inference, AMN can answer the question with high accuracy for most bAbI tasks by just looking at a single bank.

AMN presents a new design paradigm in memory networks. Unlike NTMs, where the network learns where to read/write to fine-grained address information, AMN only learns to write input entities to coarse-grained banks and entities reside within the bank. As a result, AMN is easier to train (e.g. does not require curriculum learning like NTMs) and does not require a separate sparsification mechanism like approximate nearest neighbors for inference efficiency.

AMN is timely. It has been made possible with the recent progress in dynamic networks which allows input dependent network creation and efficiency in variable sized batching as well as recent tricks in deep networks towards learning discrete decision making with high accuracy.

Apart from saving inference times, AMN can learn to reason which specific entities contribute towards the final answer improving interpretability.

---

### Decision · Program_Chairs · 2018-01-29
**ICLR 2018 Conference Acceptance Decision**

**Decision:**

Invite to Workshop Track

**Comment:**

This paper presents an interesting model which at the time of submission was still quite confusingly described to the reviewers.
A lot of improvements have been made for which I applaud the authors.
However, at this point, the original 20 babi tasks are not quite that exciting and several other models are able to fully solve them as well.
I would encourage the authors to tackle harder datasets that require reasoning or multitask settings that expand beyond babi.